# Prevention of tunneled cuffed catheter dysfunction with prophylactic use of a taurolidine urokinase lock: A randomized double-blind trial

Florence Bonkain[1,2], Jean-Claude Stolear[3], Concetta Catalano[4], Dominique Vandervelde[2], Serge Treille[5], Marie M. Couttenye[6], Annemieke Dhondt[7], Mark Libertalis[2], Mandelina Allamani[1,4], Philippe Madhoun[3], Amaryllis H. Van Craenenbroeck[6], Floris Vanommeslaeghe[7], Freya Van Hulle[1], Philippe Durieux[3], Ingrid Van Limberghen[1], Christian Tielemans[1], Karl Martin Wissing[1]*

1 Department of Nephrology, Universitair Ziekenhuis Brussel, Vrije Universteit Brussel, Brussels, Belgium, 2 Department of Nephrology, Hôpitaux Iris-Sud, Brussels, Belgium, 3 Department of Nephrology, Centre Hospitalier de Wallonie Picarde - Site IMC Tournai, Tournai, Belgium, 4 Department of Nephrology, Centre Hospitalier Universitaire Brugmann, Brussels, Belgium, 5 Department of Nephrology, Centre Hospitalier Universitaire Charleroi, Charleroi, Belgium, 6 Department of Nephrology, Universitair Ziekenhuis Antwerpen, Universiteit Antwerpen, Antwerpen, Belgium, 7 Department of Nephrology, Universitair Ziekenhuis Ghent, Universiteit Ghent, Ghent, Belgium

* karlmartin.wissing@uzbrussel.be

**Data Availability Statement:** https://doi.org/10.5281/zenodo.4514859.

## Abstract

### Background

The efficacy and cost-effectiveness of prophylactic thrombolytic locks in hemodialysis patients at high-risk of thrombotic dialysis catheter dysfunction is uncertain. We investigated this question in a double-blinded randomized controlled study.

### Methods

Prevalent hemodialysis patients from 8 Belgian hemodialysis units, with ≥2 separate episodes of thrombotic dysfunction of their tunneled cuffed catheter during the 6 months before inclusion, were randomized to either: taurolidine heparin locks thrice weekly (control arm) or the same locks twice a week combined with taurolidine urokinase locks once a week before the longest interval without HD (TaurolockU arm). The primary efficacy outcome was the incidence rate of catheter thrombotic dysfunction requiring thrombolytic locks to restore function.

### Results

68 hemodialysis patients (32 controls, 36 urokinase) were followed during 9875 catheter days between May 2015 and June 2017. Incidence rate of thrombotic catheter dysfunction was 4.8 in TaurolockU vs 12.1/1000 catheter days in control group (rate ratio 0.39; 95%CI 0.23–0.64). 15/36 (42%) catheters in the treatment group required at least one therapeutic urokinase lock vs 23/32 (72%) in the control group (P = 0.012). The two groups did not differ

**Funding:** The authors received no specific funding for this work.

**Competing interests:** The authors have declared that no competing interests exist.

significantly in catheter-related bloodstream infection and combined cost of prophylactic and therapeutic catheter locks. The TaurolockU group had a numerically higher number of episodes of refractory thrombosis.

## Conclusions

Prophylactic use of urokinase locks is highly effective in reducing the number of thrombotic catheter dysfunctions in catheters with a history of recurring dysfunction. Prophylactic use of urokinase locks did not reduce the overall costs associated with catheter locks and was associated with a numerically higher number of episodes of refractory thrombosis.

## Trial registration

ClinicalTrials.gov Identifier: NCT02036255.

## Introduction

Despite several decades of efforts promoting arteriovenous fistulas (AVF) as preferential vascular access for hemodialysis [1], tunneled cuffed catheters (TCC) remain widely used. In Belgium, 62% of incident patients with a pre-dialysis follow-up of more than 4 months nevertheless start hemodialysis (HD) with a TCC and 38% of prevalent patients continue with a catheter as long-term vascular access [2]. A recent analysis of the ERA-EDTA registry has documented increasing numbers of incident HD patients in Europe starting with a catheter and approximately 35% of patients keep a catheter as long-term vascular access [3]. Impossibility to create an AVF or graft, inadequate AVF maturation, heart failure or limited life expectancy are possible reasons that patients remain dependent on TCC as definitive vascular access [4–6]. Long-term outcomes in these patients depend on the development of cost-effective strategies to prevent TCC-related complications and prolong catheter survival.

The use of TCC is complicated by thrombotic dysfunction (TD) and less frequently by potentially life-threatening catheter-related bloodstream infection (CRBSI) [7,8]. Intraluminal locking solutions, instilled in the dead space of the TCC between successive HD sessions, are used to prevent intraluminal thrombus formation and CRBSI, but the optimal locking agent remains a matter of debate [9–11]. A meta-analysis of randomized controlled trials concluded that heparin and citrate locks have equivalent anti-thrombotic properties but that citrate was superior in preventing CRBSI [9]. The superior bactericidal effect of citrate has been documented in a Dutch randomized and controlled trial with trisodium citrate 30% locks [12] whereas it was not observed with trisodium citrate 4% locks [13–15].

Because inadvertent injection of highly concentrated trisodium citrate locks can cause serious and potentially fatal side effects [16,17], trisodium citrate 30% has been replaced with taurolidine-containing catheter locks over the last years in several European countries. Taurolidine is an antimicrobial agent showing a broad spectrum of antimicrobial activity against both bacteria and fungi without causing bacterial resistance [18,19]. It has been initially marketed in combination with citrate 4% (Taurolock™, Tauropharm GmbH, Germany), but this association showed inferior anti-thrombotic efficacy when compared with heparin in a randomized controlled trial [20]. Addition of heparin 500 IU/ml to taurolidine-citrate 4% resulted in prevention of TD equivalent to heparin 5000 IU/ml and a two-fold reduction in CRBSI [21]. The combination of taurolidine, citrate 4% and heparin 500 IU/ml (TaurolockHEP500™)

therefore provides anti-thrombotic efficacy comparable to heparin in combination with bactericidal properties similar to highly concentrated trisodium citrate.

In spite of the use of anti-thrombotic locks TCC frequently develop TD requiring either thrombolytic interventions or catheter replacement [22–24]. Refractory thrombosis results in persistent catheter dysfunction and requires TCC removal in one third of the patients [8,25]. Despite high efficacy of thrombolytic agents to dissolve the existing thrombus, recurring catheter thrombosis remains associated with worse long-term catheter survival [26].

Prophylactic use of thrombolytic locks has the potential to further improve the prevention of both TD and CRBSI. Two RCT have shown that either rt-PA (recombinant tissue plasminogen activator) once a week as compared to a standard heparin lock, or taurolidine-citrate 4% in combination with 25000 IU urokinase (Taurolock U25000™) once a week as compared to trisodium citrate 4%, resulted in significant reductions of TD and CRBSI as well as TCC removals in patients with newly inserted TCC [27,28]. Thrombotic as well as infectious complications are much more frequent during the first months after catheter placement [29]. Improved catheter survival and improved cost-effectiveness were largely driven by prevention of CRBSI in both controlled trials [27,28]. In most modern HD units however, the incidence of CRBSI in chronic patients in the maintenance phase is well below 1/1000 catheters days at risk [26,30]. In this population, the efficacy of universal primary prevention with thrombolytic locks is uncertain, and a more cost-effective approach might therefore consist in their use in catheters with a documented history of recurring TD. We therefore designed the present prospective and randomized trial comparing the efficacy and cost-effectiveness of once weekly taurolidine urokinase locks (Taurolock U25000™) against TaurolockHEP500™ in combination with placebo in prevalent TCC with at least two reversible catheter TD during the previous 6 months [31].

## Materials and methods

The study protocol and methodology have been previously published in detail [31]. This prospective randomized controlled study is reported according to the CONSORT guidelines [32] (S1 Table). This study was conducted according to the Helsinki Declaration of 1975 and approved by the central ethics committee of the Universitair Ziekenhuis Brussel (UZ Brussel) in Brussels, Belgium (B.U.N 143201523891) and by the local ethics committees in all participating centers (all in Belgium): Universitair Ziekenhuis Gent (UZ Gent) in Gent, Hôpital André Vésale (ISPPC CHU Vésale) in Montigny le Tilleul, Center Hospitalier de Wallonie Picarde (CHWapi) in Tournai, Universitair Ziekenhuis Antwerpen (UZ Antwerpen) in Antwerpen, Hôpitaux Iris Sud and CHU Brugmann both in Brussels. The study protocol has been previously registered to ClinicalTrials.gov (http://www.clinicaltrials.gov - NCT02036255).

### Study population

The study was conducted at 8 Belgian HD units between May 2015 and June 2017. Adult patients receiving chronic HD with a TCC ≥3 times a week, with a history of at least 2 separate TD treated with therapeutic urokinase (UK) lock during the previous 6 months were eligible for inclusion. Exclusion criteria were mechanical TCC dysfunction, history of heparin-induced thrombocytopenia, lower limb TCC, established intolerance to either taurolidine, citrate, heparin or UK, administration of antibiotics for ongoing TCC related infection which had ended less than one week before inclusion, or scheduled holidays >2 weeks during the 6-month study period.

## Screening

After signing written informed consent, baseline TCC function was evaluated during 3 conse-cutive screening HD sessions at least one week after the last therapeutic UK lock to confirm adequate catheter function (Fig 1). Catheter blood flow (Qb) at baseline was expressed in ml/

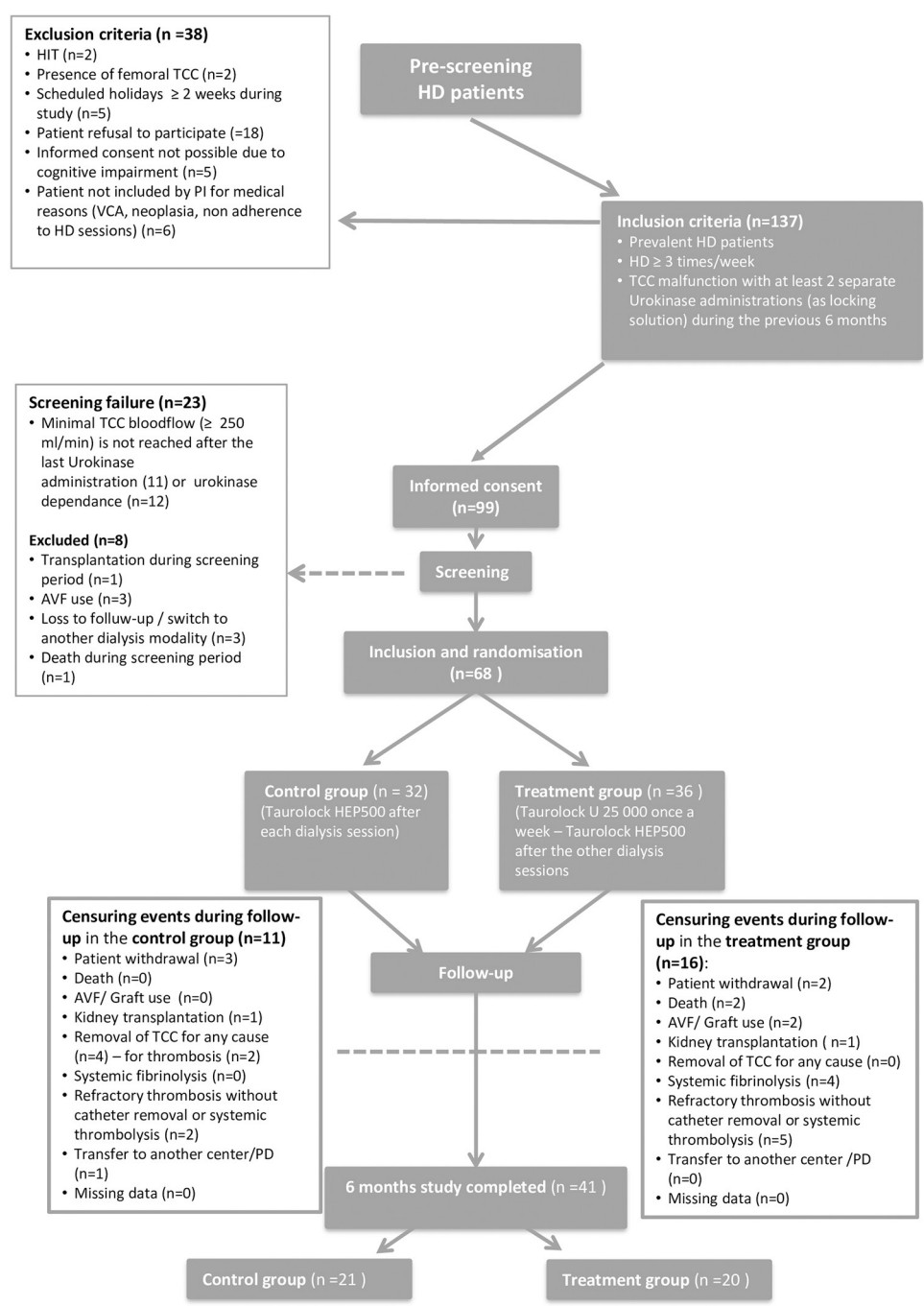

Abbreviations: HD: hemodialysis ; TCC: tunneled-cuffed catheters ; AVF: arterio-venous fistulae; PD: peritoneal dialysis; ITT: intention-to-treat

**Fig 1. Flow chart of the study.** Patients who developed a censuring event during the follow up period contributed to the study until the date of the occurrence of the censoring event.

**Table 1. Preparation and administration of study medication.**

| Lock composition | Monday Tuesday | Wednesday Thursday | Friday Saturday | Group |
|---|---|---|---|---|
| Taurolidine-citrate-heparin500 | X | X | X[1] | |
| Placebo | | | X[2] | TaurolockHep500 (control group) |
| Taurolidine-citrate-heparin500 | X | X | | |
| Taurolidine-citrate | | | X[2] | TaurolockU |
| Urokinase 25000 U | | | X[2] | |

[1] Taurolidine-citrate-heparin500 and Taurolidine-citrate used for the preparation of catheter locks after the last dialysis session of the week were packaged in indistinguishable vials with full blinding of staff, investigators and patients.

[2] Urokinase and the corresponding placebo were packaged in indistinguishable vials.

min and calculated by dividing total processed blood volume by treatment time. To be eligible for randomization the following criteria had to be met at each screening HD session: blood flow > 250 ml/min, pre-pump pressure > -250 mmHg and post-pump pressure < +250 mmHg.

## Study medication allocation

Patients were allocated using block randomization with permuted blocks of four stratified for the participating centers [31]. All patients in the study received TaurolockHEP500 locks after the two first hemodialysis sessions of the week (Table 1). Patients allocated to the intervention group received Taurolock U25000™ locks after the third HD session of the week (before the longest HD interval) whereas the control group received TaurolockHEP500 with full blinding (Table 1) [31]. Each catheter lock was instilled into the TCC lumen at a volume equivalent to the dead space of the TCC (range 1.5 to 2.2 ml for arterial lumen and 1.6 to 2.4 ml for venous lumen).

The locks administered after the last dialysis session of the week were prepared by dissolving placebo powder in Taurolidine-citrate-heparin500 and urokinase 25000 U powder in Taurolidine-citrate.

## Definition and management of catheter-related thrombotic dysfunction

Catheter TD was the main outcome criterion of the study and defined by either: 1) a drop of mean TCC blood flow during two consecutive HD sessions by >15% compared to baseline TCC blood flow (despite reversal of catheter lines) or 2) a complete occlusion of one or both catheter lines impeding HD.

In case of TD, therapeutic UK was administered by dissolving a vial of 100 000 IU UK in a volume corresponding to the dead space of the TCC (dose ranging from 21 740 to 32 260 IU/ml). It was administered either at the start of the session if HD could not be started or continued, or as lock at the end of the HD session (instead of the usual lock). Per protocol, 3 consecutive therapeutic UK locks were systematically administered for each episode of TD.

## Primary outcome

Primary efficacy outcome was the median number of catheter TD requiring therapeutic UK locks at the end of the follow-up.

## Secondary outcomes

Refractory thrombosis was defined as TCC blood flow not restored above the threshold for UK use after 3 therapeutic UK locks. According to local practice in participating centers

TCC with refractory thrombosis could be either replaced or treated by systemic fibrinolysis. Additional secondary outcomes were CRBSI, TCC removal for any cause and HD adequacy determined monthly by single-pool Kt/V [33]. The overall weekly costs of the locks were compared in both groups per patient during the 6-month period, taking into account that the control group received TaurolockHEP500 thrice a week (3 euros per vial) and treatment group received TaurolockU (15 euros per vial) once a week plus TaurolockHEP500 twice a week. The cost of prophylactic locks in each group were added to the cost of therapeutic locks corresponding to 3 consecutive 100000 IU UK vials at 64 euros per dose for each episode of TD.

## Sample size calculation and statistical analysis

Based on a retrospective review of patients with recurring catheter TD in one of the participating centers we expected a median of 2 UK treatments over 6 months (range 1–5) in the TaurolockHEP500 arm [31]. Sample size calculation for rank data with ties, a type I error of 5% and power equal to 90%, indicated that 25 patients per group were required to detect a reduction by 50% in therapeutic urokinase use [31]. Anticipating a dropout rate of approximately 20% a total recruitment of 64 (32 per group) was planned.

Data were unblinded after completion and verification of the source data collection. The study database was frozen after unblinding treatment allocation and no further data were entered or corrected. All analyses were done according to the intention-to-treat principle.

The null hypothesis of no difference in the number of catheter TD was tested by the Wilcoxon rank sum test. To account for possible differences in time at risk due to TCC removal and patient dropout in the two arms of the study, the primary outcome was also expressed as a rate (number of TCC dysfunctions requiring UK treatment/ 1000 catheter days at risk) with 95% confidence intervals. Time at risk was defined from inclusion to either end of the 6-month follow-up period or the occurrence of a censoring event (refractory TD, catheter removal, change of HD center, use of AVF, shift to peritoneal dialysis, kidney transplantation or patient death). Individual patients could therefore contribute multiple episodes of TD during follow up. The effect of the intervention was expressed by the rate ratio with 95% confidence interval. Survival free of the primary endpoint was estimated by the Kaplan-Meier method with hypothesis testing using the log-rank test. No multivariate analysis was foreseen in the initial data analysis plan of this prospective intervention trial. Because time on dialysis before inclusion differed more than expected by chance between the two treatment arms reviewers requested to adjust the primary efficacy endpoint for this variable. The adjusted incidence rate of thrombotic dysfunction was calculated by bivariate Poisson regression with dialysis vintage as additional predictor variable in the analysis. Time to first thrombotic catheter dysfunction was modeled by Cox proportional hazard regression with dialysis vintage as an additional predictor variable. The proportional hazards assumption was tested by graphically displaying log-log plots of survival probability using the STATA stphplot command for quartiles of dialysis vintage and by Schönfeld residual for dialysis vintage as a continuous variable with the STATA estat phtest command. All summary statistics and hypothesis testing was realized on the available dataset without removal of outliers. Missing values were not imputed. All hypothesis testing were done using two-tailed tests with a type 1 error of 0.05 as threshold for significance to reject the null hypothesis. The anonymized dataset and STATA do file of the analysis are publicly available in a data repository (https://doi.org/10.5281/zenodo.4514859).

All statistical analysis were realized with STATA version 15.1 (StataCorp, College Station, USA).

## Results

### Study population

A total of 137 HD patients fulfilled the inclusion criteria of having received at least 2 separate UK locks for TD in the previous 6 months and were screened for participation as detailed in Fig 1. Finally, 68 patients were randomized (36 patients in the TaurolockU group and 32 patients in the control group) and followed during 9875 catheter days (5017 days for the TaurolockU group and 4858 days for the control group).

### Baseline characteristics

Both groups were comparable regarding age, sex, comorbidities, cause of end-stage renal disease and anticoagulant or anti-platelet therapy (Table 2). Patients receiving TaurolockHep500 thrice a week had a longer HD vintage than patients in the TaurolockU group (median of 4.4 years with IQR 1.8–8.3 vs 2.7 years with IQR 1.1–4.5 years respectively; P = 0.01). All other characteristics (TCC age, number of TCC removals for thrombosis and TCC performance determined by mean blood flow during the screening week, URR and Kt/V) did not differ more than expected by chance between both groups.

### Primary outcome

Patients receiving TaurolockU once a week needed significantly less therapeutic UK locks than patients receiving TaurolockHEP500: 24 vs 59 therapeutic UK locks respectively, P = 0.0012 by Wilcoxon rank sum test (Table 3). IR of TD was 4.8/1000 catheter days in the TaurolockU group and 12.1/1000 catheter days in the control group (IR ratio 0.39–95% CI 0.23–0.64; P<0.0001). As the control arm had a significantly longer dialysis vintage as compared to the TaurolockU arm, the incidence rate ratio was adjusted for this parameter by Poisson regression modelling. The univariate IR ratio for TaurolockU as only variable in the model was 0.39 (95%CI 0.24 to 0.63; P<0.0001). Longer dialysis vintage was associated with a significantly lower incidence of thrombotic dysfunction (IR ratio 0.93 per year (95%CI 0.87 to 0.99; P = 0.04). After adjustment for dialysis vintage the adjusted IR ratio for TaurolockU use was 0.34 (95%CI 0.21 to 0.56; P<0.0001). The Kaplan-Meier estimate of catheter survival free of TD was significantly better in the TaurolockU group (Fig 2; P = 0.01). The univariate Cox proportional hazards ratio or thrombotic catheter dysfunction for the use of TaurolockU was 0.43 (95%CI 0.22 to 0.84; P = 0.013). After adjustment for dialysis vintage in the Cox model the hazards ratio was 0.37 (95%CI 0.18 to 0.74; P = 0.005). The difference in therapeutic urokinase lock was due to a lower number of reductions of blood flow by >15% on two consecutive sessions (2.6/1000 catheter days in the TaurolockU group vs 9.5/1000 catheter days in the TaurolockHEP500 arm; IR ratio 0.27 (95% CI 0.14 to 0.52); P<0.0001). However, prophylactic use of thrombolytic locks did not reduce the incidence of TD with complete occlusion of the catheter and impossibility to start dialysis therapy (Table 3).

### Secondary outcomes

In patients receiving TaurolockU once a week, 9 episodes of TD met the predefined criteria for the secondary outcome of refractory thrombosis (Table 3), leading to systemic thrombolytic therapy with UK in 4 cases. In the 5 remaining patients, TCC were left in place without additional treatment by the treating nephrologists, in spite of incomplete reversal of TD, because the patients were deemed to have either a contra-indication for systemic fibrinolysis or no other possibility for TCC replacement. In the control group, 2 TCC with persisting dysfunction were replaced for refractory thrombosis and 2 others were left in place without other

**Table 2. Patient characteristics.**

| Characteristic | TaurolockU N = 36 | TaurolockHEP N = 32 | P-value[2] |
|---|---|---|---|
| Age–yr (mean ± SD) | 72.3 ± 13.8 | 73.2 ± 12.2 | 0.794 |
| Female sex–n (%) | 16 (44.4) | 19 (59.4) | 0.236 |
| Cause of end-stage renal disease–n (%) | | | 0.370 |
| Diabetic nephropathy | 11 (30.5) | 9 (28.1) | |
| Hypertensive or vascular nephropathy | 19 (52.8) | 10 (31.3) | |
| Glomerulonephritis | 1 (2.8) | 3 (9.4) | |
| Polycystic renal disease | 1 (2.8) | 3 (9.4) | |
| Tubulo-interstitial disease | 1 (2.8) | 2 (6.3) | |
| Unknown | 3 (8.3) | 5 (15.6) | |
| Duration of dialysis–yr (median, IQR) | 2.7 (1.1–4.5) | 4.40 (1.8–8.3) | 0.01 |
| Catheter (last) vintage—days (median, IQR) | 542 (231–1277) | 533 (263–921.5) | 0.716 |
| Catheter localization–n (%) | | | 0.470 |
| Right internal jugular vein | 29 (80.6) | 23 (71.9) | |
| Left internal jugular vein | 7 (19.4) | 8 (25.0) | |
| Subclavian vein | 0 (0) | 1 (3.1) | |
| History of catheter removal for thrombosis–n (%) | 10 (27.8) | 9 (30.0) | 0.606 |
| Number of catheters removed—n (%) | | | 0.454 |
| 1 | 5 (50.0) | 7 (77.8) | |
| 2 | 3 (30.0) | 1 (11.1) | |
| 3 | 0 (0) | 1 (11.1) | |
| 4 | 1 (10.0) | 0 (0) | |
| 5 | 1 (10.0) | 0 (0) | |
| History of AVF/AVG failure due to thrombosis | 6 | 5 | 0.925 |
| Number of episodes—n (%) | | | |
| 1 | 4 (11.1) | 5 (15.6) | |
| 2 | 1 (2.8) | 0 (0) | |
| 3 | 1 (2.8) | 0 (0) | |
| Number of urokinase locks per patient in the 6m before inclusion–(median; 95% CI) | 3.06 (2.34–3.77) | 3.63 (2.82–4.43) | 0.283 |
| Comorbidity–n (%)[1] | | | |
| Diabetes | 18 (50.0) | 14 (43.8) | 0.606 |
| Arterial hypertension | 33 (91.7) | 28 (87.5) | 0.699 |
| Peripheral vascular disease | 10 (27.8) | 6 (18.8) | 0.408 |
| Ischemic cardiomyopathy | 15 (41.7) | 10 (31.3) | 0.374 |
| Deep venous thrombosis | 1 (2.8) | 3 (9.4) | 0.336 |
| Pulmonary embolism | 2 (5.6) | 1 (3.1) | 1 |
| Active neoplasia | 4 (11.1) | 1 (3.1) | 0.360 |
| Medications at inclusion–n (%) | | | |
| Aspirin | 21 (58.3) | 17 (53.1) | 0.666 |
| Clopidogrel | 2 (5.6) | 2 (6.3) | 1 |
| Warfarin | 6 (16.7) | 7 (21.9) | 0.759 |
| Low molecular weight heparin | 2 (5.6) | 1 (3.1) | 1 |
| Anticoagulation during hemodialysis—n (%) | | | 0.524 |
| Anticoagulation free hemodialysis | 0 (0) | 1 (3.1) | |
| Unfractionated heparin during hemodialysis | 22 (61.1) | 22 (68.8) | |
| Low molecular weight heparin during hemodialysis | 11 (30.6) | 6 (18.8) | |
| Other | 3 (8.3) | 3 (9.4) | |
| Hemodialysis modality–n (%) | | | 0.865 |

*(Continued)*

**Table 2.** (Continued)

| Characteristic | TaurolockU N = 36 | TaurolockHEP N = 32 | P-value[2] |
|---|---|---|---|
| Hemodialysis high flux | 30 (83.3) | 26 (81.3) | |
| Hemodiafiltration (post-dilution) | 6 (16.7) | 5 (15.6) | |
| Hemodiafiltration (mid-dilution) | 0 (0) | 1 (3.1) | |
| Hemodialysis duration—min (median, smallest and largest time) | 240 (180–240) | 240 (210–240) | 0.118 |
| Locking solution at inclusion | | | 1 |
| Taurolock HEP500 | 34 (94.4) | 30 (93.8) | |
| Sodium citrate | 1 (2.8) | 1 (3.1) | |
| Heparin | 1 (2.8) | 1 (3.1) | |
| Baseline catheter blood flow (mean ± SD; ml/min) | 300±23 | 297±26 | 0.61 |
| Serum albumin (mean ± SD; g/l) | 36.5±4.5 | 36.7±5.2 | 0.89 |
| Baseline URR at inclusion (mean ± SD; %) | 72.4 ±7.9 | 73.5±9.5 | 0.60 |
| Baseline Kt/V at inclusion (mean ± SD) | 1.45±0.3 | 1.49±0.3 | 0.56 |
| Platelet count–(mean ± SD; x10³/l) | 211.6±86.2 | 209.8±85.1 | 0.93 |
| Hemoglobin level (mean ± SD; g/dl) | 10.7±1.1 | 10.5±1.2 | 0.49 |
| C-reactive protein (mean ± SD; mg/l) | 12±14.1 | 10.7±15.1 | 0.71 |
| EPO dosis–IU/Kg/week (median, 95% CI) | 81.5 (0–129.5) | 74 (0–138) | 0.96 |

Abbreviations: SD, standard deviation; IQR, interquartile range; AVF, arterio-venous fistula; AVG, arterio-venous graft; URR, ureum reduction rate.

[1] Some comorbidities may be additive in the same patient.

[2] Due to randomization P values are not provided to test the null hypothesis but to inform whether groups differed by chance more than expected based on a type I error of 5%.

**Table 3. Primary and secondary efficacy outcomes.**

| Outcomes | TaurolockU N = 36 | TaurolockHEP N = 32 | P-value |
|---|---|---|---|
| **Primary outcome** | | | |
| Total number of thrombotic dysfunctions (TD) | 24 | 59 | 0.0012[1] |
| Incidence rate (N/1000 catheter days) | 4.8 | 12.3 | |
| Incidence rate ratio (95% CI) | 0.39 (0.23–0.64) | | <0.0001 |
| TD with reduction of blood flow[2] | 13 | 46 | |
| Incidence rate (N/1000 catheter days) | 2.6 | 9.5 | |
| Incidence rate ratio (95% CI) | 0.27 (0.14–0.51) | | <0.0001 |
| TD with impossibility to start dialysis therapy[3] | 11 | 13 | |
| Incidence rate (N/1000 catheter days) | 2.2 | 2.7 | |
| Incidence rate ratio (95% CI) | 0.82 (0.33–1.98) | | 0.32 |
| **Secondary outcomes** | | | |
| Refractory thrombosis (total)[4] | 9 (25%) | 4 (12.5%) | 0.23 |
| Catheter removal for refractory thrombosis | 0 | 2 | 0.218 |
| Systemic fibrinolysis | 4 | 0 | 0.116 |
| Refractory thrombosis (without catheter removal or fibrinolysis) | 5 | 2 | 0.434 |
| Catheter-related bloodstream infection | 1 | 0 | 1 |

[1] Test of null hypothesis of equal number of urokinase locks by Wilcoxon rank sum test.

[2] Reduction of mean blood flow by >15% of baseline value during screening during two consecutive dialysis sessions. Therapeutic urokinase lock administered at end of dialysis session.

[3] Complete catheter occlusion with impossibility to start hemodialysis session. Therapeutic urokinase lock administered before the start of the hemodialysis session.

[4] Median time to refractory thrombosis: TaurolockU 2.7 (IQR 1.0 to 3.6) months and TaurolockHEP 3.1 (IQR 1.2 to 4.8) months; P = 0.75.

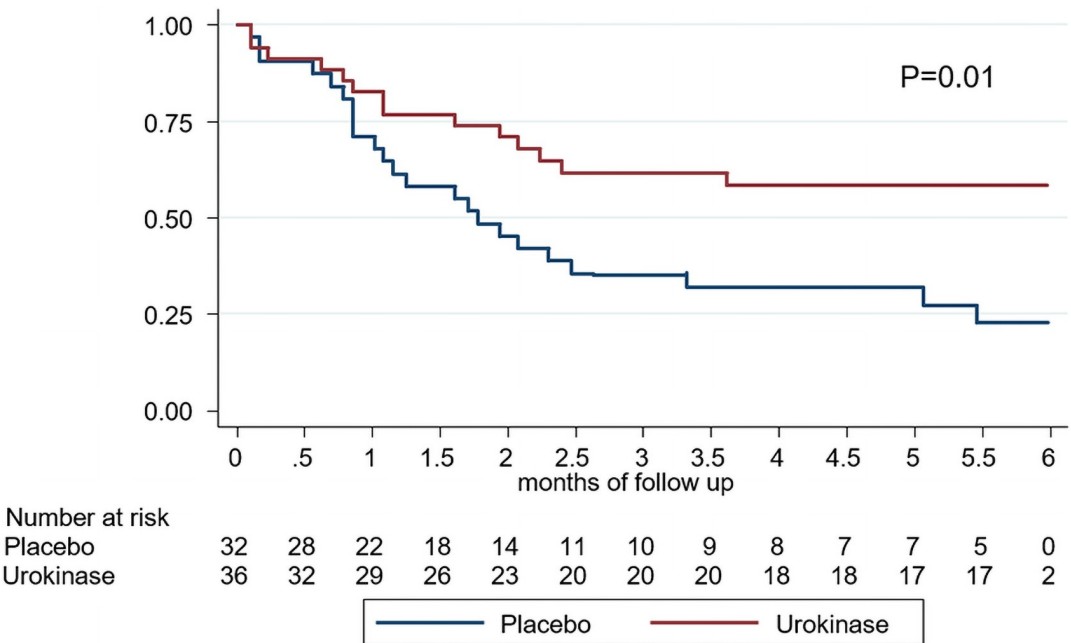

**Fig 2. Proportion of catheters free of thrombotic dysfunction.** Kaplan-Meier estimates with censoring at the time of the first thrombotic dysfunction. Population at risk during the 6 months follow up is indicated under the graph. Placebo (blue lines) and once weekly urokinase (red lines). P value of null hypothesis of no difference in survival by log-rank test = 0.01.

therapeutic action (Table 3). The only CRBSI of the whole cohort was observed in the Tauro-lockU group.

Seven patients in each group terminated the study before the end of 6-month follow up period for reasons other than refractory TD (Table 4; P = 0.111). Therefore, a total of 20 patients (55.6%) in the TaurolockU group and 21 patients (65.6%) in the control group completed the 6-month study period with a functional TCC (P = 0.626).

## Hemodialysis adequacy

There was no difference in HD adequacy parameters in the intervention group compared to patients receiving TaurolockHEP500 thrice weekly (Table 5). Hemodialysis efficacy remained stable in both groups over time.

**Table 4. Reasons for early study termination other than refractory thrombosis.**

| Reason | TaurolockU | TaurolockHEP |
|---|---|---|
| Catheter damage[1] | 0 | 2 |
| Loss to follow-up | 0 | 1 |
| Decision patient / PI | 2 | 3 |
| Kidney transplantation | 1 | 1 |
| AVF use | 2 | 0 |
| Death | 2 | 0 |
| Total | 7/36 (19.4%) | 7/32 (21.9%) |

[1] Loss of mechanical integrity by fissuring.

**Table 5. Evolution of hemodialysis adequacy parameters during the 6-month study period.**

| Biological parameters (mean ± SD) | TaurolockU N = 36[1] | TaurolockHEP N = 32 |
|---|---|---|
| **Urea reduction ratio (URR)** | | |
| Month 1 | 72.0 ± 10.1 | 69.3 ± 13.7 |
| Month 2 | 72.2 ± 9.9 | 72.6 ± 8.5 |
| Month 3 | 72.2 ± 10.8 | 72.5 ± 8.9 |
| Month 4 | 71.2 ± 8.6 | 72.5 ± 8.6 |
| Month 5 | 72.7 ± 7.3 | 72.3 ± 10.3 |
| Month 6 | 73.9 ± 9.2 | 73.8 ± 8.8[2] |
| **Kt/V** | | |
| Month 1 | 1.49 ± 0.38 | 1.44 ± 0.39 |
| Month 2 | 1.50 ± 0.43 | 1.47 ± 0.30 |
| Month 3 | 1.46 ± 0.36 | 1.45 ± 0.28 |
| Month 4 | 1.41 ± 0.34 | 1.46 ± 0.29 |
| Month 5 | 1.51 ± 0.31 | 1.42 ± 0.27 |
| Month 6 | 1.51 ± 0.39 | 1.47 ± 0.27[3] |

[1] Number of patients with assessments of dialysis efficacy during months 1–6 in both groups: TaurolockU: 36, 31, 28, 26, 21, 15; TaurolockHEP: 31, 29, 26, 24, 21, 20.

[2] P = 0.95.

[3] P = 0.70.

## Adverse events

No severe bleeding or bleeding related deaths occurred in both groups. Two deaths from infectious complications unrelated to TCC occurred in the TaurolockU group.

## Cost effectiveness

Prophylactic use of TaurolockU was associated with a significant increase in the median cost per patient for the 6-month study period of prophylactic locks (525 euros; IQR 278–546) compared to TaurolockHEP500 (207 euros; IQR 160–216; P<0.0001). Nevertheless, total cost for both prophylactic and therapeutic catheter locks, taking into account the price of therapeutic UK, did not differ significantly: 546 euros (IQR 481.5–547.5) in the TaurolockU group vs 554 euros (IQR 234–781) in the TaurolockHEP500 group (Table 6).

**Table 6. Costs of prophylactic and therapeutic catheter locks per patient during the 6-month study period [1].**

| | Treatment group | Control group | P- value[3] |
|---|---|---|---|
| TaurolockHEP500[2] | 150 (79.3–156) | 207 (160.1–216) | NA[4] |
| TaurolockU[2] | 375 (198.2–390) | | NA[4] |
| Total cost prophylaxis | 525 (277.5–546) | 207 (160.1–216) | <0.0001 |
| Therapeutic urokinase locks | 0 (0–192) | 384 (0–576) | 0.0012 |
| | Range (0–768) | Range (0–1344) | |
| Total cost | 546 (481.5–547.5) | 554.4 (234–781.1) | 0.96 |
| Total cost/week[5] | 21 (21–33.4) | 23.1 (9–30.3) | 0.35 |

[1] Cost expressed in Euros per patient for the 6 month study period (median with IQR).

[2] TaurolockHEP500: taurolidine heparin locking solution; TaurolockU: taurolidine urokinase locking solution.

[3] P values for the null hypothesis of no difference in price calculated by the Wilcoxon rank sum test.

[4] NA: not applicable; as groups differed in the number of TaurolockHEP500 and TaurolockU catheter locks per protocol no hypothesis testing was done.

[5] Price of prophylactic and therapeutic locks per week calculated by dividing total cost by the number of week patients were on study medication.

## Discussion

The efficacy of prophylactic thrombolytic locks to prevent dialysis catheter dysfunction has been first demonstrated in HD patients with newly inserted TCC in 2011 [27]. In this study, the use of rt-PA once a week was associated with a drastic reduction in catheter TD and CRBSI when compared with heparin locks three times a week [27]. The same beneficial effect on TD, CRBSI and catheter survival was also reported in a prospective and randomized study comparing a regimen identical to our intervention arm to citrate 4% catheter locks [28]. Increased survival of newly placed catheters in these studies has been largely driven by a significant reduction in CRBSI [27,28]. However, the incidence of thrombotic and infectious complications has been shown to be 5–10 times higher during the first 3–6 months after TCC placement [29].

The present multicenter randomized double-blinded study evaluates the impact of prophylactic thrombolytic locks in HD patients with prevalent catheters, used in their majority for more than 1 year. The study recruited patients with a history of recurring episodes of TD to target a population with a potential to attain cost-effectiveness beyond the first months after catheter placement. The results clearly show that in this high-risk population prophylactic use of taurolidine UK locks once a week reduces the use of therapeutic urokinase locks for thrombotic dysfunction by 61%. This effect is completely in line with 62% reduction in rescue Alteplase use by TaurolockU in incident TCC as compared to Citrate 4% locks [28] and the 61% reduction in thrombotic dysfunction when rt-PA locks were added to citrate 4% locks in a cohort of patients with prevalent TCC [30].

Use of thrombolytic locks in newly placed TCC improves catheter survival mainly by reducing the incidence rate of CRBSI [27,28]. The risk of both CRBSI and non-infectious complications decreases rapidly during the months after placement of TCC [29]. Contrary to the previously reported benefits in patients with incident TCC, use of prophylactic UK locks in the present controlled trial including prevalent catheters with a history of recurring thrombotic dysfunction did not reduce the risk of irreversible catheter dysfunction due to refractory thrombosis. This result is completely in line with the results of a Canadian multi-center cohort study comparing the effect of weekly rt-PA prophylactic locks against citrate 4% locks in patients at high risk of TCC TD or with a history of CRBSI [30]. In this cohort rt-PA locks reduced the use of thrombolytic locks for TD by 61%, an effect identical to TaurolockU in the present study, without significant effect on catheter stripping or removal as measures of irreversible catheter dysfunction [30].

Surprisingly, in the present study TaurolockU was even associated with a numerically higher incidence rate for the composite endpoint of refractory TD. The study was not adequately powered to investigate catheter survival and the difference between the two groups might have been the result of random variability. However, a two-fold higher incidence of refractory thrombosis in the TaurolockU arm must raise the question whether this difference in outcomes was causally related to the different use of thrombolytic locks in the two groups. As TaurolockU is unlikely to increase the risk of thrombotic dysfunction, the much more frequent administration of therapeutic UK locks in the control arm might be a possible explanation. As three therapeutic UK locks were administered for each episode of thrombotic dysfunction catheters in the control arm received approximately 100 therapeutic UK locks more than in the TaurolockU arm. As therapeutic locks have a four-fold higher urokinase concentration than TaurolockU, the control arm might have benefitted from more frequent administration of locks with potentially better efficacy on extraluminal clot and fibrin sheath formation.

The present multicenter study had an overall incidence rate of CRBSI of 0.1/1000 catheter days, which is considerably lower than reported in most previous studies [9]. A post hoc cost-

benefit analysis of prophylactic rt-PA locking solutions in newly-placed TCC concluded that increased treatment-related costs were offset by a reduction in hospitalizations for CRBSI and that the degree of cost-effectiveness was a function of the background incidence of CRBSI [34]. Winnicki et al also showed that the overall costs, including expenses in relation with the treatment of CRBSI and catheter dysfunction, were 43% lower for patients with newly-placed catheters receiving prophylactic locks with TaurolockU as compared to citrate 4% [28]. On the contrary, in prevalent catheters with a low incidence of CRBSI, once-weekly prophylactic use of rt-PA had no effect on the incidence of CRBSI and was associated with significantly higher treatment costs [30].

These observations highlight the potentially essential role of timing in the use of thrombo-lytic locks to prevent TCC complications. The proven efficacy in reducing the risk of CRBSI in newly placed catheters provides a rationale to use thrombolytic locks for the first 3–6 months after TCC placement, when the risk of infectious complications is highest [27–29]. After this period, and in the absence of conclusive data on improving catheter survival and CRBSI, a more selective use in patients with recurring TD appears as the most rational and cost-effective approach. A recent prospective and randomized study from Qatar investigated Tauroloc-kHep500 and TaurolockU locks at the end of the third hemodialysis of the week in an unse-lected cohort of prevalent and incident (45%) TCC [35]. TaurolockU significantly reduced the combined outcome of catheter loss by thrombosis and infection, although the effect on each of the two composite outcomes did not reach significance. The authors did not investigate whether the treatment effect differed between incident and prevalent TCC, but this study sug-gests that TaurolockU could improve catheter survival in settings where prevalent TCC have a higher incidence of CRBSI than in the present cohort.

Our study is the first double-blinded randomized controlled study assessing the efficacy of taurolidine UK locks in prevalent TCC at high risk of recurring catheter TD. Although the double-blinded allocation considerably reduces the possibility for investigator bias in the assessment of the outcomes, this study has several limitations. First, the selected target popula-tion limits generalization of the results to other HD populations. However, most of the cur-rently available evidence shows that thrombolytic locks cause a similar reduction in the relative risk of TD by about 60% [28,30,35]. The background risk of the target population is therefore likely to have an impact mainly on the absolute risk reduction. Selection of patients for this treatment will therefore depend on the need to balance efficacy and cost in the local setting.

The lower limit of dialysis catheter blood flow for inclusion into the present study was >250 ml/min, which is lower than the 300 ml/min limit used to define adequate dialysis cathe-ter function in KDOQI guidelines [36]. However, the 300 ml/min threshold is not based on high-quality evidence but expert opinion and it has been shown that the majority of TCC with a Qb between 250 and 300 ml/min provide adequate dialysis [37]. As half of the participants in the present study had a baseline Qb below 300 ml/min, our inclusion criteria reflect current clinical practice and the fact that nephrologists are clearly reluctant to rapidly replace TCC in patients with recurrent thrombotic dysfunction.

Third, our primary outcome was partly based on measurement of TCC blood flow, which can be considered as only a surrogate marker for thrombotic dysfunction. The drop in Qb to be used as cut-off for the diagnosis of thrombotic dysfunction is subject to discussion. The reduction by >15% of baseline Qb followed the expert opinion that a decline of >10% in Qb at constant prepump pressure is highly suggestive of impending access dysfunction [38]. This threshold was also in line with routine clinical practice in the different participating centers. This reduction in flow is lower than in other intervention trials of thrombotic dysfunction [28] but the Qb drop of >15% had to be observed during 2 consecutive HD sessions to reduce the

risk of false endpoint attribution in case of reduced blood flow not related to TD. In addition, treatment allocation was fully blinded to avoid investigator bias in the adjudication of the efficacy endpoints of the study.

Fourth, in spite of randomization the control group had a longer time on dialysis therapy at the moment of inclusion into the study. However, longer dialysis vintage was associated with a lower risk of thrombotic dysfunction. This variable therefore acted as a negative confounder of the protective effect and adjustment for dialysis time further lowered the incidence rate ratio of thrombotic catheter dysfunction with the use of TaurolockU. Fifth, the cost analysis of the intervention was limited to the cost of the two catheter locks and of therapeutic urokinase locks. This unfortunately does not take into account additional cost such as the time nurses spent to treat occluded or dysfunctional catheters. Finally, our study is limited by the relative small sample size. Power calculation were based on the reduction in episodes of thrombotic dysfunction resulting in the administration of thrombolytic urokinase locks. The sample size was indeed sufficient to detect a highly significant effect on the primary efficacy endpoint with a relatively narrow confidence interval. However, the sample size provides insufficient power to detect effects on clinically relevant outcomes such as refractory thrombosis and TCC replacement. It is therefore impossible to assess whether the observed increase in refractory thrombosis in the TaurolockU arm is the result of random variability or due to the more frequent use of high-dose therapeutic urokinase locks in the control arm.

## Conclusions

In conclusion, prophylactic use of taurolidine UK locks once a week in a high-risk HD population with a history of multiple TD reduced the number of episodes of thrombotic TCC dysfunction requiring therapeutic UK by approximately 60% as compared to taurolidine heparin locks thrice weekly. The intervention was neutral in terms of cost as the lower cost due to avoidance of therapeutic locks was compensated by the higher cost of taurolidine UK prophylactic locks. We observed numerically less episodes of refractory thrombotic dysfunction in the control arm, which might be the result of the more frequent use of high dose therapeutic urokinase locks in these catheters.

## Supporting information

**S1 File. CONSORT 2010 checklist.** CONSORT 2010 checklist of information to include when reporting a randomized trial.
(DOC)

**S2 File. Protocol-taurolockFINAL.**
(PDF)

## Acknowledgments

The authors would like to acknowledge all the HD and research nurses of the participating centers without whom this study would not have been possible. We also thank Nils Noppe from the pharmacy of the Universitair Ziekenhuis Brussel who processed the randomization and sent the study medication to the participating sites.

## Author Contributions

**Conceptualization:** Florence Bonkain, Christian Tielemans, Karl Martin Wissing.

**Data curation:** Florence Bonkain, Freya Van Hulle, Ingrid Van Limberghen, Karl Martin Wissing.

**Formal analysis:** Florence Bonkain, Karl Martin Wissing.

**Investigation:** Florence Bonkain, Jean-Claude Stolear, Concetta Catalano, Dominique Vandervelde, Serge Treille, Marie M. Couttenye, Annemieke Dhondt, Mark Libertalis, Mandelina Allamani, Philippe Madhoun, Amaryllis H. Van Craenenbroeck, Floris Vanommeslaeghe, Freya Van Hulle, Philippe Durieux, Christian Tielemans.

**Methodology:** Florence Bonkain, Christian Tielemans, Karl Martin Wissing.

**Project administration:** Florence Bonkain, Concetta Catalano, Ingrid Van Limberghen, Christian Tielemans.

**Resources:** Ingrid Van Limberghen, Christian Tielemans.

**Software:** Ingrid Van Limberghen.

**Supervision:** Christian Tielemans.

**Validation:** Jean-Claude Stolear, Concetta Catalano, Dominique Vandervelde, Serge Treille, Marie M. Couttenye, Annemieke Dhondt, Mark Libertalis, Mandelina Allamani, Philippe Madhoun, Amaryllis H. Van Craenenbroeck, Floris Vanommeslaeghe, Freya Van Hulle, Philippe Durieux, Ingrid Van Limberghen, Christian Tielemans, Karl Martin Wissing.

**Visualization:** Jean-Claude Stolear.

**Writing – original draft:** Florence Bonkain, Jean-Claude Stolear, Concetta Catalano, Dominique Vandervelde, Serge Treille, Marie M. Couttenye, Annemieke Dhondt, Mark Libertalis, Mandelina Allamani, Philippe Madhoun, Amaryllis H. Van Craenenbroeck, Floris Vanommeslaeghe, Freya Van Hulle, Philippe Durieux, Christian Tielemans, Karl Martin Wissing.

**Writing – review & editing:** Florence Bonkain, Jean-Claude Stolear, Concetta Catalano, Dominique Vandervelde, Serge Treille, Marie M. Couttenye, Annemieke Dhondt, Mark Libertalis, Mandelina Allamani, Philippe Madhoun, Amaryllis H. Van Craenenbroeck, Floris Vanommeslaeghe, Freya Van Hulle, Philippe Durieux, Ingrid Van Limberghen, Christian Tielemans, Karl Martin Wissing.

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
