## [Decision Letter · Decision Letter 0]

29 Dec 2020

PONE-D-20-30158

Prevention of tunneled cuffed catheter dysfunction with prophylactic use of a taurolidine urokinase locks: a prospective and randomized double-blind trial

PLOS ONE

Dear Dr. Wissing

Thank you for submitting your manuscript to PLOS ONE. After careful consideration, we feel that it has merit but does not fully meet PLOS ONE’s publication criteria as it currently stands. Therefore, we invite you to submit a revised version of the manuscript that addresses the points raised during the review process.

We look forward to receiving your revised manuscript.

Kind regards,

Pasqual Barretti, Ph.D., MD

Academic Editor

PLOS ONE

Additional Editor Comments:

The manuscript is interesting, well written and appropriate in relation to its methodology. However, reviewers 1 and 3 made important suggestions, which can greatly improve the final wording. So, my option is major revision

2. During your revisions, please note that a simple title correction is required: the title in the manuscript file describes "use of a taurolidine urokinase lock", whereas the submission site describes "use of a tarolidine urokinase locks". Please update the online submission information to remove the "s" from "lock".

3. To comply with PLOS ONE submission guidelines, in your Methods section, please provide additional information regarding your statistical analyses. For more information on PLOS ONE's expectations for statistical reporting, please see https://journals.plos.org/plosone/s/submission-guidelines.#loc-statistical-reporting.

4. Thank you for including your ethics statement:  This study was conducted according to the Helsinki Declaration of 1975 and approved by the central ethic committee of the Universitair Ziekenhuis Brussel (UZ Brussel) in Brussels, Belgium and each local participating ethic committee (B.U.N 143201523891)"

Reviewers' comments:

Reviewer's Responses to Questions

**Comments to the Author**

1. Is the manuscript technically sound, and do the data support the conclusions?

Reviewer #1: Yes

Reviewer #2: Yes

Reviewer #3: Yes

2. Has the statistical analysis been performed appropriately and rigorously? 

Reviewer #1: Yes

Reviewer #2: Yes

Reviewer #3: Yes

3. Have the authors made all data underlying the findings in their manuscript fully available?

Reviewer #1: Yes

Reviewer #2: Yes

Reviewer #3: Yes

4. Is the manuscript presented in an intelligible fashion and written in standard English?

Reviewer #1: Yes

Reviewer #2: Yes

Reviewer #3: Yes

5. Review Comments to the Author

Reviewer #1: I have reviewed carefully the manuscript entitled “Prevention of tunneled cuffed catheter dysfunction with prophylactic use of a taurolidine urokinase locks: a prospective and randomized double-blind trial” (PONE-D-20-30158) by Florence Bonkain et al.

This multicentered, double-blinded randomized controlled study was conducted enrolling 68 hemodialysis patients (36 patients in the TaurolockU group and 32

patients in the control group) from eight Belgian hemodialysis units between May 2015 and June 2017.

After a follow-up of 9875 catheter days (5017 days for the TaurolockU group and 4858 days for the control group), the study demonstrated that prophylactic use of urokinase locks once per week is highly effective in reducing the number of thrombotic catheter dysfunctions without increasing total cost for patients with a history of recurring catheter dysfunction.

Generally speaking, this manuscript is a well-prepared work.

The study's theme is intriguing and clinically important.

The English writing, including the expression, fluency, and readability of this manuscript is good.

However, I have several minor comments

[Title]

# Is it necessary to add a term “prospective” in front of a “randomized double-blind trial?”

[Abstract]

# Some necessary components (ex: the source and number of participants, study place and period…etc.) are lacking in the abstract section

[Results]

# The control group has significantly longer “duration of dialysis” than the intervention group ((median, IQR) , 4.40 (1.8-8.3) versus 2.7 (1.1-4.5), p=0.01). I suggest that the authors make comment regarding the association between “duration of dialysis” and “dysfunction of catheter” in Discussion section.

Reviewer #2: The authors successfully define the study outcomes and gap in literature that their study addresses, with a a clear methodology, appropriate statistical analysis, and conclusions based on the data. Primary outcome, its measurement, time points at which measurements were taken for the primary outcome, and the analysis metric were all satisfactory according to the CONSORT criteria.

Sample size calculations were included taking in to account predicted rate of attrition. Recruitment and justification of the number of participants was satisfactory. The authors provide a reference to their published study protocol. Allocations within each arm of the study were done in a method which could not be predicted.

Statistical analysis supports the conclusions of the paper both within primary and secondary outcomes with statistical tests named and rationale described, as well as explaining cost-effectiveness between the 2 interventions was not significant in the context of the study and its implementation. The lack of adverse events was clearly addressed by the authors.

The authors have submitted a well-designed study which satisfies the CONSORT criteria, as well as satisfying its primary outcome measure. The null hypothesis can be rejected as the authors demonstrate that prophylactic use of taurolidine uorkinase lock in the defined group reduces the number

of thrombotic TCC dysfunction compared to taurolidine heparin locks, taking into account of other factors which may influence the outcomes.

It is the opinion of this reviewer that the study is suitable for publication.

Reviewer #3: This is a nice paper. The data are very interesting and relevant.

I have three concerns:

1. The statistical analysis performed do not take acount the misbalance between groups in regard of duration of dialysis. To this is required a generalized linear model.

2. In a cohort in which outcome may occur at many points of time, there is a need for Cox analysis regression that compares the hazard risk of outcome (catheter thrombosis), take account the time interval between the time zero to the event, and make survival curves adjusted to the misbalances between groups. In this case, bivariate analysis will be sufficient because groups were heterogeneous only in regard of the duration of dialysis.

2. Author should take into account the cost of waste of time to make thrombolysis in a catheter in the cost analysis.

6. PLOS authors have the option to publish the peer review history of their article (what does this mean?). If published, this will include your full peer review and any attached files.

Reviewer #1: **Yes: **Chih-Chung Shiao

Reviewer #2: **Yes: **Habib Akbani

Reviewer #3: No

---

## [Author Response · Author response to Decision Letter 0]

17 Feb 2021

Reviewer’s comments

Reviewer 1

Comment reviewer 1:

# Is it necessary to add a term “prospective” in front of a “randomized double-blind trial?”

Answer:

We agree with the remark of reviewer 1 and deleted the term “prospective” in the title of the manuscript. 

Comment reviewer 1:

# Some necessary components (ex: the source and number of participants, study place and period…etc.) are lacking in the abstract section

Answer:

We added all the requested information in the abstract section (see page 2).

Comment reviewer 1:

I suggest that the authors make comment regarding the association between “duration of dialysis” and “dysfunction of catheter” in Discussion section.

Answer:

Reviewer 1 is right to say there was a significantly longer hemodialysis vintage in the control group as compared with the TaurolockU group. This point has also been raised by reviewer three. We refer to the answers and additional analysis discussed in response to comments 1 and 2 below.

At the suggestion of reviewer 1 the potential role of dialysis vintage as a confounder of the protective effect of TaurolockU on catheter dysfunction has now been added to the discussion. (Page 26)

Reviewer 2

Answer:

We thank reviewer 2 for the positive assessment of our study.

Reviewer 3 

I have three concerns:

Comment Reviewer 3:

1. The statistical analysis performed do not take account the misbalance between groups in regard of duration of dialysis. To this is required a generalized linear model.

Answer (answer with details of the statistical analysis is part of the pdf document of the revision):

Reviewer 3 is right that the two groups differ in terms of duration of dialysis in spite of randomization. This is not unusual when multiple baseline characteristics are tested as on average 1 out of 20 is likely to differ “significantly” with a type I error fixed at 5%.

At the suggestion of the reviewer, we have tested whether dialysis vintage might have acted as a potential confounder of the effect of urokinase locks on the occurrence of thrombotic dysfunction. In our paper the effect of urokinase locks was tested by two types of analysis: 1) the incidence rate of thrombotic dysfunction, considering multiple episodes of thrombotic dysfunction per catheter under study and 2) the time to the first occurrence of thrombotic dysfunction in Kaplan-Meier estimates of survival free of thrombotic dysfunction. In the survival analysis catheters were censored after the first occurrence of thrombotic dysfunction. 

Multivariate analysis of the incidence rate ratio after adjustment for dialysis vintage was done by Poisson regression modelling and the hazards ratio in time to event regression was realized by Cox proportional hazards regression (this part is described under question 2 of the reviewer).

The univariate IRR calculated with the STATA “poisson” command was 0.39 and thus identical the 0.39 IRR when calculating point estimates and confidence intervals for the incidence-rate ratio with the STATA “ir” command. 

When dialysis vintage was introduced as a continuous variable into the model it became clear that longer dialysis vintage was associated with significantly lower incidence rates of thrombotic dysfunction. The longer dialysis vintage in the control arm therefore tended to act as a negative confounder of the association between urokinase locks and thrombotic catheter dysfunction.

After adjustment for this effect the IRR was reduced to 0.34 and remained statistically highly significant.

We have conducted the same analysis by introducing 4 quartiles of dialysis vintage as dummy variables into the regression model. 

The adjusted estimate of the IRR of urokinase versus placebo was 0.32.

There was no difference in the fit of the two models based on the likelihood ratio test but slightly better fit of the model with hemodialysis vintage as a continuous data was suggested by lower values of the Akaike's and Bayesian information criterions.

We have therefore added the following phrases to the results section: “As the control arm had a significantly longer dialysis vintage as compared to the TaurolockU arm, the incidence rate ratio was adjusted for this parameter by Poisson regression modelling. The univariate IR ratio for TaurolockU as only variable in the model was 0.39 (95%CI 0.24 to 0.63; P<0.0001). Longer dialysis vintage was associated with a significantly lower incidence of thrombotic dysfunction (IR ratio 0.93 per year (95%CI 0.87 to 0.99; P=0.04). After adjustment for dialysis vintage the adjusted IR ratio for TaurolockU use was 0.34 (95%CI 0.21 to 0.56; P<0.0001).” (Page 16)

Comment Reviewer 3:

2. In a cohort in which outcome may occur at many points of time, there is a need for Cox analysis regression that compares the hazard risk of outcome (catheter thrombosis), take account the time interval between the time zero to the event, and make survival curves adjusted to the misbalances between groups. In this case, bivariate analysis will be sufficient because groups were heterogeneous only in regard of the duration of dialysis.

Answer:

The use of urokinase was associated with a univariate Cox hazard ratio for thrombotic catheter dysfunction of 0.43 (95%CI 0.22 to 0.84). 

Adding dialysis vintage as a continuous variable reduced the hazards ratio to 0.37.

Adding hemodialysis vintage as dummy variables of quartiles reduced the hazards ratio to 0.31. 

The proportional hazards assumption was respected for the two models. The likelihood of the two models did not differ significantly when tested by the likelihood ratio test. The model with hemodialysis vintage as a continuous data was chosen because it had slightly lower values of the Akaike's and Bayesian information criterions.

The following phrase was therefore added to the paper: “The univariate Cox proportional hazards ratio for the use of urokinase locks was 0.43 (95%CI 0.22 to 0.84; P=0.013). When dialysis vintage was added to the Cox model the hazards ratio was 0.37 (95%CI 0.18 to 0.74; P=0.005).” (Page 16)

We are grateful to reviewer 3 to have raised the issue of a possible confounding role of dialysis vintage on the reduction of thrombotic dysfunction by TaurolockU locks. Following the suggestion of reviewer 1 this has now been added in the discussion to the limitations of the paper. 

"Fourth, in spite of randomization the control group had a longer time on dialysis therapy at the moment of inclusion into the study. However, longer dialysis vintage was associated with a lower risk of thrombotic dysfunction. This variable therefore acted as a negative confounder of the protective effect and adjustment for dialysis time further lowered the incidence rate ratio of thrombotic catheter dysfunction with the use of TaurolockU." (Page 26)

Comment Reviewer 3: 

2. Author should take into account the cost of waste of time to make thrombolysis in a catheter in the cost analysis.

Answer:

This is a correct criticism. We did not record the time nurses spent on treating dysfunctional catheters which includes application of therapeutic locks but also flushing of catheters and aspiration of thrombi. This shortcoming has been added to the discussion.

"Fifth, the cost analysis of the intervention was limited to the cost of the two prophylactic catheter locks and of therapeutic urokinase locks. This unfortunately does not take into account additional cost such as the time nurses spent to treat occluded or dysfunctional catheters."

We consider that the revisions in response to the comments of the reviewers have improved the quality of our paper and hope that it can be considered at present suitable for publication in PlosOne.

---

## [Decision Letter · Decision Letter 1]

4 May 2021

Prevention of tunneled cuffed catheter dysfunction with prophylactic use of a taurolidine urokinase lock: a randomized double-blind trial

PONE-D-20-30158R1

Dear Dr. Wissing, 

We’re pleased to inform you that your manuscript has been judged scientifically suitable for publication and will be formally accepted for publication once it meets all outstanding technical requirements.

Kind regards,

Pasqual Barretti, Ph.D., MD

Academic Editor

PLOS ONE

Additional Editor Comments (optional):

The manuscript can be accepted for publication, But, I strongly recommend the authors do the suggestion from the reviewer 4> Table 5 - is footnote 3 still required?

Figure 2 - is 1 patient missing from the figure? The N at risk are 32 and 35 for the groups - should be 36?

Reviewers' comments:

Reviewer's Responses to Questions

**Comments to the Author**

1. If the authors have adequately addressed your comments raised in a previous round of review and you feel that this manuscript is now acceptable for publication, you may indicate that here to bypass the “Comments to the Author” section, enter your conflict of interest statement in the “Confidential to Editor” section, and submit your "Accept" recommendation.

Reviewer #2: All comments have been addressed

Reviewer #3: All comments have been addressed

Reviewer #4: (No Response)

2. Is the manuscript technically sound, and do the data support the conclusions?

Reviewer #2: Yes

Reviewer #3: Yes

Reviewer #4: (No Response)

3. Has the statistical analysis been performed appropriately and rigorously? 

Reviewer #2: Yes

Reviewer #3: Yes

Reviewer #4: Yes

4. Have the authors made all data underlying the findings in their manuscript fully available?

Reviewer #2: Yes

Reviewer #3: Yes

Reviewer #4: Yes

5. Is the manuscript presented in an intelligible fashion and written in standard English?

Reviewer #2: Yes

Reviewer #3: Yes

Reviewer #4: Yes

6. Review Comments to the Author

Reviewer #2: I thank my fellow reviewers for suggesting improvements upon the original manuscript. It is clear that the authors have done an excellent job in addressing their concerns, and I would like to in particular draw attention to the robust detail of statistical analysis that has been completed. My previous comments remain:

The authors successfully define the study outcomes and gap in the literature that their study addresses, with a clear methodology, appropriate statistical analysis, and conclusions based on the data. The primary outcome, its measurement, time points at which measurements were taken for the primary outcome, and the analysis metric were all satisfactory according to the CONSORT criteria.

Sample size calculations were included taking into account the predicted rate of attrition. Recruitment and justification of the number of participants were satisfactory. The authors provide a reference to their published study protocol. Allocations within each arm of the study were done in a method that could not be predicted.

The statistical analysis supports the conclusions of the paper both within primary and secondary outcomes with statistical tests named and rationale described, as well as explaining cost-effectiveness between the 2 interventions was not significant in the context of the study and its implementation. The lack of adverse events was clearly addressed by the authors.

The authors have submitted a well-designed study that satisfies the CONSORT criteria, as well as satisfying its primary outcome measure. The null hypothesis can be rejected as the authors demonstrate that prophylactic use of taurolidine UK in the defined group reduces the number

of thrombotic TCC dysfunction compared to taurolidine heparin locks, taking into account other factors which may influence the outcomes.

It is the opinion of this reviewer this manuscript remains suitable for publication and I thank the authors for making appropriate changes.

Reviewer #3: All questions were completely adressed. I want to greet the authors by the nice study. The answers were sactisfatory an the changes were made.

Reviewer #4: Table 5 - is footnote 3 still required?

Figure 2 - is 1 patient missing from the figure? The N at risk are 32 and 35 for the groups - should be 36?

7. PLOS authors have the option to publish the peer review history of their article (what does this mean?). If published, this will include your full peer review and any attached files.

Reviewer #2: **Yes: **Habib Akbani

Reviewer #3: No

Reviewer #4: No

---

## [Editor Report · Acceptance letter]

11 May 2021

PONE-D-20-30158R1 

Prevention of tunneled cuffed catheter dysfunction with prophylactic use of a taurolidine urokinase lock: a randomized double-blind trial 

Dear Dr. Wissing:

I'm pleased to inform you that your manuscript has been deemed suitable for publication in PLOS ONE. Congratulations! Your manuscript is now with our production department. 

Kind regards, 

on behalf of

Prof. Pasqual Barretti 

Academic Editor

PLOS ONE